# Noise Attention Learning: Enhancing Noise Robustness by Gradient Scaling

**Yangdi Lu**
Department of Computing and Software
McMaster University
`luy100@mcmaster.ca`

**Yang Bo**
Department of Computing and Software
McMaster University
`boy2@mcmaster.ca`

**Wenbo He**
Department of Computing and Software
McMaster University
`hew11@mcmaster.ca`

## Abstract

Machine learning has been highly successful in data-driven applications but is often hampered when the data contains noise, especially label noise. When trained on noisy labels, deep neural networks tend to fit all noisy labels, resulting in poor generalization. To handle this problem, a common idea is to force the model to fit only clean samples rather than mislabeled ones. In this paper, we propose a simple yet effective method that automatically distinguishes the mislabeled samples and prevents the model from memorizing them, named Noise Attention Learning. In our method, we introduce an attention branch to produce attention weights based on representations of samples. This attention branch is learned to divide the samples according to the predictive power in their representations. We design the corresponding loss function that incorporates the attention weights for training the model without affecting the original learning direction. Empirical results show that most of the mislabeled samples yield significantly lower weights than the clean ones. Furthermore, our theoretical analysis shows that the gradients of training samples are dynamically scaled by the attention weights, implicitly preventing memorization of the mislabeled samples. Experimental results on two benchmarks (CIFAR-10 and CIFAR-100) with simulated label noise and three real-world noisy datasets (ANIMAL-10N, Clothing1M and Webvision) demonstrate that our approach outperforms state-of-the-art methods.

## 1   Introduction

With the emergence of highly-curated datasets such as ImageNet [1] and CIFAR [2], deep neural networks (DNNs) have achieved remarkable performance on many classification tasks. However, in real-world applications, it is extremely time-consuming and expensive, sometimes even impossible to label a new large-scale dataset containing fully correct annotations. To alleviate this problem, one may obtain the data with lower quality annotations efficiently through online keywords queries [3] or crowdsourcing [4], but *noisy labels* (e.g. a cat image is mislabeled as dog) are inevitably introduced consequently. Previous study [5] has empirically demonstrated that noisy labels in training data are problematic for deep neural networks, resulting in overfitting and performance degradation. Therefore, designing robust algorithms against noisy labels is of great practical importance.

Given the training data consists of clean (correctly labeled) samples and mislabeled samples, DNNs have been observed to correctly predict the true labels for all training samples during a *early learning*

36th Conference on Neural Information Processing Systems (NeurIPS 2022).

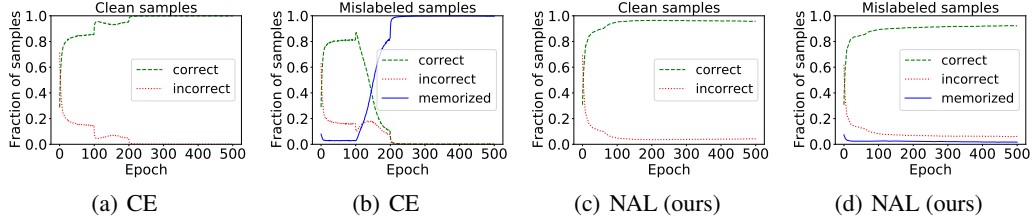

(a) CE       (b) CE       (c) NAL (ours)       (d) NAL (ours)

Figure 1: The results of training a ResNet34 [14] with cross entropy (CE) loss and our proposed method on the CIFAR-10 with 40% symmetric noise. Plots (a) and (c) show the fraction of clean samples that are predicted correctly (green) and incorrectly (red). In contrast, plots (b) and (d) show the fraction of mislabeled samples that are predicted correctly (green), memorized (i.e. the prediction equals the wrong label, shown in blue), and incorrectly predicted as neither the true nor the labeled class (red). For clean samples, both models predict them correctly with the increasing of epochs. However, for mislabeled samples in plot (b), the model trained with CE loss first predicts the true labels correctly, but eventually memorizes all mislabeled samples. In contrast, our approach prevents memorization of mislabeled samples, allowing the model to continue learning the clean samples to attain high accuracy on all samples.

stage, and then start to make incorrect predictions as it gradually *memorizes* the mislabeled samples [6, 7] (see Figure 1(a) and 1(b)). Based on this empirical finding, many existing approaches [8–11] use the training loss or prediction confidence to identify mislabeled samples in the early learning stage, i.e., the samples with small loss or high confidence are more likely to be clean samples. After dividing the samples into clean and mislabeled sets, one can simply train the model only based on clean set [8, 11], or correct these mislabeled samples by pseudo-labels [9, 12], or treat them as unlabeled samples and train the model using semi-supervised learning algorithms [10]. However, these methods typically involve sophisticated training procedure modifications and sometimes require high computational costs.

In this paper, we propose Noise Attention Learning (NAL) which enjoys simplicity and universality from a new perspective. The core idea behind NAL is to let the model automatically differentiate the mislabeled samples from clean samples, and prevent the memorization of mislabeled samples in training. Since the memorization of DNNs has a preference for easy (clean) samples, the predictive power of a sample's representation aligns with its label cleanliness [6]. To transform this qualitative observation into a quantitive measure, we introduce an attention branch that outputs scalar attention weights to indicate the predictive power of representations. We incorporate the attention weights in our loss function, so that the attention branch can be learned to divide samples, i.e., clean samples are learned to be associated with large weights, while mislabeled ones with small weights. In this way, the proposed method does not require extra steps (e.g. modeling loss distribution [9, 10], selecting small-loss samples [8]) to separate the training set, effectively simplifying the training procedure. Theoretically, we demonstrate that NAL neutralizes the effect of the mislabeled samples on the gradient and ensures the contribution of clean samples to the whole gradient remains dominant, thereby preventing the deep neural networks from overfitting mislabeled samples.

The proposed method has a similar effect on gradients as an existing regularization method ELR [7]. In general, the performance of pure regularization methods cannot be compared to complex methods. Despite that the regularization methods can effectively hinder the memorization of mislabeled samples, the limited number of clean samples makes the model not rich enough to generalize effectively to the held-out data. To solve this problem, ELR uses temporal ensembling [13] to estimate the targets in its loss function to correctly guide the model for improving performance. Similarly, instead of directly using noisy labels as target distribution in our loss function, we adopt SELC [12] to estimate the targets, allowing the generation of more clean labels for learning sufficiency.

Empirically, we show that the proposed approach achieves better robustness than the state-of-the-art methods on two benchmarks (CIFAR-10 and CIFAR-100 [2]) with simulated label noise and three challenging datasets in noisy label literature (ANIMAL-10N [15], Clothing1M [16] and Webvision [3]) with real-world label noise. To better understand our method, we conduct many empirical analyses, including memorization analysis, gradient analysis, attention weights, quality of targets, hyperparameter sensitivity and ablation study, to verify our theoretical explanations and design goals.

## 2 Preliminaries

**Classification with Noisy Labels.** Consider the $K$-class classification problem in noisy-label scenario, the ground truth label $y$ is unobservable. We only have a noisy training set $\hat{D} = \{(\boldsymbol{x}^{[i]}, \hat{y}^{[i]})\}_{i=1}^{N}$, where $\boldsymbol{x}^{[i]}$ is an input and $\hat{y}^{[i]} \in \mathcal{Y} = \{1, \ldots, K\}$ is the corresponding noisy label. We denote $\hat{\boldsymbol{y}}^{[i]} \in \{0,1\}^K$ as one-hot vector of noisy label $\hat{y}^{[i]}$. A DNN $\mathcal{N}_\theta$ maps an input $\boldsymbol{x}^{[i]}$ to a $K$-dimensional logits and then feeds the logits to a softmax function to obtain $\boldsymbol{p}^{[i]}$ of the conditional probability of each class. $\theta$ denotes the parameters of the DNN and $\boldsymbol{z}^{[i]} \in \mathbb{R}^{K \times 1}$ denotes the logits (i.e. pre-softmax output). We refer to it as prediction branch in this work. We have $\boldsymbol{z}^{[i]} = \mathcal{N}_\theta(\boldsymbol{x}^{[i]})$ and $\boldsymbol{p}^{[i]} = \texttt{softmax}(\boldsymbol{z}^{[i]})$. Without knowing the ground truth joint probability distribution $P(\boldsymbol{x}, y)$, the typical cross-entropy (CE) loss is often used as empirical risk to measure how well the model fits the training set $\hat{D}$ as follows:

$$\mathcal{L}_{\text{ce}} = \frac{1}{N} \sum_{i=1}^{N} \ell_{\text{ce}}(\hat{\boldsymbol{y}}^{[i]}, \boldsymbol{p}^{[i]}) = -\frac{1}{N} \sum_{i=1}^{N} (\hat{\boldsymbol{y}}^{[i]})^\top \log(\boldsymbol{p}^{[i]}). \tag{1}$$

**Early Learning Phenomenon** When optimizing $\mathcal{L}_{\text{ce}}$ by stochastic gradient descent (SGD), it has been observed that DNNs fit the easy (clean) samples first before memorizing of hard (mislabeled) samples [6]. Figure 1(a) and 1(b) shows this qualitative behavior. The model begins by learning to predict the true labels, even for majority of mislabeled samples, but eventually learns to predicts wrong labels due to memorization of mislabeled samples.

**Label Noise Model.** The generation of real-world label noise is unpredictable, a typical methodology to dealing with noisy labels is to posit a noise model and develop robust algorithms under the hypothetical noise model. The algorithms are then tested on the real-world noisy datasets to see how effective they are. A widely-accepted noise model is class-conditional noise [17, 18], where label noise is independent of input features and true labels are corrupted by a *symmetric* or *asymmetric* noise transition matrix (simulation details are in Appendix B.2). Instance-dependent noise [19, 20] is another noise model in which the noise is affected not only by the label but also by the input feature.

## 3 Noise Attention Learning

In this section, we present our approach NAL for learning with noisy labels. NAL consists of three key elements: (1) An attention branch based on learned representations to produce the attention weight for each training sample; (2) A noise attention loss is specifically designed to learn the attention weights; (3) A target estimation strategy to help generate more clean labels for sufficient learning.

### 3.1 Attention Branch based on Learned Representations

To enable the deep neural networks to produce an attention weight that reflects the quality of learned representation, we introduce an attention branch just after the penultimate layer of the original model. We denote $M$-dimensional penultimate layer (representation) of input $\boldsymbol{x}^{[i]}$ as $H^{[i]}$. $H^{[i]}$ is shared in both prediction and attention branches. For each input, the prediction branch outputs the softmax prediction $\boldsymbol{p}^{[i]}$ as usual. The attention branch contains one fully connected layer to produce a single scalar value $h^{[i]}$, and $\texttt{sigmoid}$ function is applied to scale it between 0 to 1. Specifically, we have $h^{[i]} = W H^{[i]} + b$, where $W \in \mathbb{R}^{1 \times M}$ denotes the weights and $b \in \mathbb{R}$ denotes the bias in penultimate layer of the attention branch. We have the attention weight $\tau^{[i]}$ for each sample $\boldsymbol{x}^{[i]}$ as

$$\tau^{[i]} = \texttt{sigmoid}(h^{[i]}), \quad \tau^{[i]} \in (0, 1). \tag{2}$$

Note that $W$ and $b$ are extra network parameters need to be learned to produce meaningful $\tau$'s. Therefore, attention weights are not fixed but dynamic during the training.

### 3.2 Noise Attention Loss

The early learning phenomenon reveals that the DNNs memorize the clean samples before the mislabeled samples. Therefore, clean samples are more likely to have better learned representations than mislabeled samples in the early learning phase. To enable the attention weights to automatically

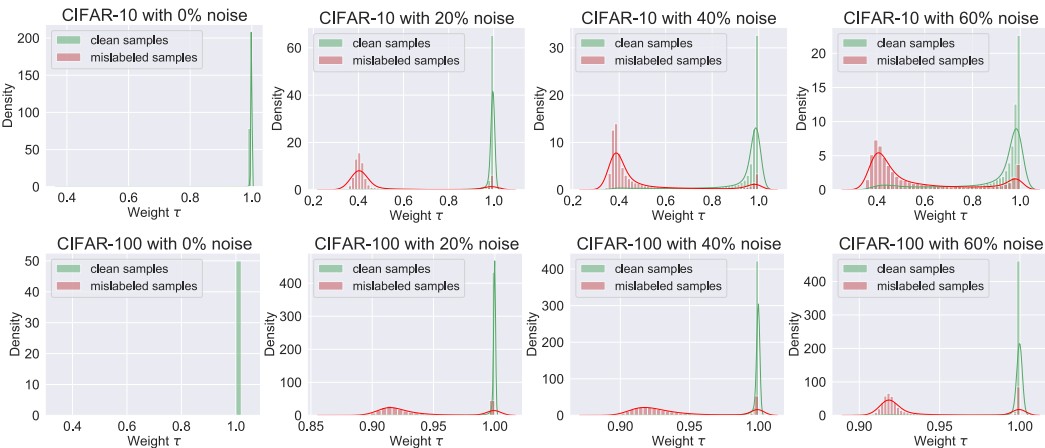

Figure 2: The distribution of attention weights on the CIFAR-10/CIFAR-100 with different ratios of symmetric label noise using ResNet34.

capture the difference of representations as well as learn the classification task, we propose a noise attention loss consisting of two terms: attention term $\mathcal{L}_{\mathrm{a}}$ and boost term $\mathcal{L}_{\mathrm{b}}$.

$$\mathcal{L}_{\mathrm{NAL}} = \mathcal{L}_{\mathrm{a}} + \lambda\mathcal{L}_{\mathrm{b}} = -\frac{1}{N}\sum_{i=1}^{N}(\hat{\boldsymbol{y}}^{[i]})^{\top}\log\big(\underbrace{\tau^{[i]}(\boldsymbol{p}^{[i]} - \hat{\boldsymbol{y}}^{[i]}) + \hat{\boldsymbol{y}}^{[i]}}_{\text{perceptual shortcut}}\big) - \frac{\lambda}{N}\sum_{i=1}^{N}\log(\tau^{[i]}), \quad (3)$$

where $\lambda$ is the hyperparameter. Intuitively, the attention term $\mathcal{L}_{\mathrm{a}}$ can be interpreted as a dynamic expansion of the CE loss, where the prediction is modified to 'perceptual shortcut' $\tau(\boldsymbol{p} - \hat{\boldsymbol{y}}) + \hat{\boldsymbol{y}}$ (Note that many existing methods [9, 21, 22] use this linear interpolation formula to infer the label distribution, while the proposed NAL applies it to prediction distribution for learning the meaningful attention weights). Assuming no representations are corrupted and all weights $\tau \to 1$, the attention term $\mathcal{L}_{\mathrm{a}}$ reduces to exactly $\mathcal{L}_{\mathrm{ce}}$ in Eq. (1). On the contrary, when all $\tau \to 0$, the attention term $\mathcal{L}_{\mathrm{a}}$ becomes "lazy" as it produces many zero gradients.

The mechanism of attention term can be explained as follows: (1) For mislabeled samples, the model does not memorize them in the early learning phase. Therefore, their representations are under learned compared to clean samples and $(\boldsymbol{p} - \hat{\boldsymbol{y}})$ remains large. By minimizing $\mathcal{L}_{\mathrm{a}}$, it forces $\tau$ of mislabeled samples toward 0, which is treated as 'shortcut'. (2) For clean samples, the model memorizes them first, resulting in $(\boldsymbol{p} - \hat{\boldsymbol{y}}) \to \boldsymbol{0}$. It makes $\tau$ have no influence on minimizing $\mathcal{L}_{\mathrm{a}}$ as the prediction distribution is already equal to $\hat{\boldsymbol{y}}$. As a result, by solely minimizing $\mathcal{L}_{\mathrm{a}}$, we may obtain a trivial optimization result that the model always produces $\tau \to 0$ for any inputs. To avoid this 'lazy' learning circumstance, we have another boost term $\mathcal{L}_{\mathrm{b}}$. It can be interpreted as a binary CE loss, where the target of $\tau$ is always 1 for all inputs. By adding the boost term, the attention weights of clean samples are pushed to 1.0, and weights of mislabeled samples are still close to 0 as expected.

To verify the effectiveness of $\mathcal{L}_{\mathrm{NAL}}$ in discriminating the mislabeled samples from clean samples, we empirically analyze the weight distribution of clean and mislabeled samples with simulated symmetric noise. Figure 2 shows results on CIFAR-10 and CIFAR-100 with different ratios of symmetric noise. In 0% noise cases, we observe that all samples have weights close to 1.0, which indicates that $\mathcal{L}_{\mathrm{NAL}}$ reduced to exactly $\mathcal{L}_{\mathrm{ce}}$. Thus, the proposed loss would not affect the performance when training with clean data. At other ratios of label noise, there is a clear separation between attention weights of clean and mislabeled samples. The weights of clean samples form a spike close to 1.0, whereas the weights of most mislabeled samples are plainly less than 1.0, which satisfies our design goal.

### 3.3 Target Estimation

For regularization methods, directly using the noisy labels as target distribution in loss function is less effective as the limited number of clean samples makes the model not rich enough to generalize effectively to the held-out data, especially at the high noise level. To yield better performance, ELR [7]

---
**Algorithm 1** Noise Attention Learning (NAL) pseudocode
---
1: **Input:** DNN $\mathcal{N}_\theta$; Training set $\hat{D} = \{(\boldsymbol{x}^{[i]}, \hat{\boldsymbol{y}}^{[i]})\}_{i=1}^N$; Hyperparameters $\lambda$ and momentum $\alpha$; Total epoch $E$.
2: **for** epoch $e = 1$ **to** $E$ **do**
3:     **for** each minibatch $B$ **do**
4:         **for** $i$ in $B$ **do**
5:             **Obtain** $\boldsymbol{z}^{[i]}, h^{[i]} = \mathcal{N}_\theta(\boldsymbol{x}^{[i]})$.
6:             **Obtain** $\boldsymbol{p}^{[i]}, \tau^{[i]} = \texttt{softmax}(\boldsymbol{z}^{[i]}), \texttt{sigmoid}(h^{[i]})$.
7:             **Update** $\boldsymbol{t}_{[e]}^{[i]}$ by Eq. (4).
8:         **end for**
9:         **Calculate** $\mathcal{L}_{\text{NAL}} = -\frac{1}{|B|}\sum_{i=1}^{|B|}(\boldsymbol{t}_{[e]}^{[i]})^\top \log\left(\tau^{[i]}(\boldsymbol{p}^{[i]} - \boldsymbol{t}_{[e]}^{[i]}) + \boldsymbol{t}_{[e]}^{[i]}\right) - \frac{\lambda}{|B|}\sum_{i=1}^{|B|}\log(\tau^{[i]})$.
10:         **Update** $\theta$ using stochastic gradient descent.
11:     **end for**
12: **end for**
13: **Output:** $\theta$.
---

use the temporal ensembling [13] to estimate targets in its regularizer and steer the model towards these new targets. The ensemble predictions are formed by an exponential moving average of historical outputs. Specifically, in the $m$-th epoch, it can be expressed as $\tilde{\boldsymbol{p}}_{[m]} = \sum_{j=1}^m (1-\alpha)\alpha^{m-j}\boldsymbol{p}_{[j]}$, where $0 \le \alpha < 1$ is the momentum and $\boldsymbol{p}_{[j]}$ represents the output of $j$-th epoch. In this paper, we use the ensemble strategy in SELC [12] to estimate our targets $\boldsymbol{t}$ for the $m$-th epoch as follows:

$$\boldsymbol{t}_{[m]} = \underbrace{\alpha^m \hat{\boldsymbol{y}}}_{\text{initial term}} + \underbrace{\sum_{j=1}^m (1-\alpha)\alpha^{m-j}\boldsymbol{p}_{[j]}}_{\text{ensemble term}}. \tag{4}$$

The initial term preserves the original noisy labels with exponential decaying weights $\alpha^m$. In the early learning phase (small $m$), the overall target relies on noisy labels to learn the model. As the learning continues (larger $m$), $\alpha^m$ approaches 0. The overall target eventually depends on the ensemble term $\tilde{\boldsymbol{p}}_{[m]}$ and penalizes model predictions that are inconsistent with this target. In our experiment, we will demonstrate that estimated targets are more accurate than original noisy labels. We simply replace noisy label $\hat{\boldsymbol{y}}$ with ensemble target $\boldsymbol{t}$ in our loss. Algorithm 1 shows pseudocode of our method.

## 4 Theoretical Justification

In this section, we demonstrate the noise robustness of $\mathcal{L}_{\text{NAL}}$ by analyzing how it scales the gradients accordingly to prevent memorization of mislabeled samples. The omitted proofs in this section are in Appendix A. For clarity of explanation, we denote the true label of sample $\boldsymbol{x}$ as $y \in \{1, ..., K\}$. The ground-truth distribution over labels for sample $\boldsymbol{x}$ is $q(y|\boldsymbol{x})$, and $\sum_{k=1}^K q(k|\boldsymbol{x}) = 1$. Consider the case of a single ground-truth label $y$, then $q(y|\boldsymbol{x}) = 1$ and $q(k|\boldsymbol{x}) = 0$ for all $k \neq y$. We denote the prediction probability as $p(k|\boldsymbol{x})$ and $\sum_{k=1}^K p(k|\boldsymbol{x}) = 1$. For notation simplicity, we denote $p_k$, $q_k$, $p_y$, $q_y$, $p_j$, $q_j$ as abbreviations for $p(k|\boldsymbol{x})$, $q(k|\boldsymbol{x})$, $p(y|\boldsymbol{x})$, $q(y|\boldsymbol{x})$, $p(j|\boldsymbol{x})$ and $q(j|\boldsymbol{x})$, where $j$ represents $j$-th entry. First, we show how the CE loss fails in noisy-label scenario.

**Lemma 1.** *Given the cross-entropy loss $\mathcal{L}_{ce}$ in Eq. (1), we rewrite the sample-wise loss $\ell_{ce} = -\sum_{k=1}^K q_k \log p_k$. Its gradient with respect to $z_j$ is*

$$\frac{\partial \ell_{ce}}{\partial z_j} = \begin{cases} p_j - 1 \le 0, & q_j = q_y = 1 \text{ (}j\text{ is the true class for } \boldsymbol{x}\text{)}, \\ p_j \ge 0, & q_j = 0 \text{ (}j\text{ is not the true class for } \boldsymbol{x}\text{)}, \end{cases} \tag{5}$$

*where $z_j$ is the $j$-th entry of logits $\boldsymbol{z}$. $q_j = q_y = 1$ means $j$ equals the true class $y$.*

In clean-label scenario, Lemma 1 ensures that, during SGD, learning direction of CE loss continues towards the true class as the corresponding gradient remains negative. However, in noisy-label scenario, the learning direction fluctuates. Suppose $j$ is true class and equals $y$, but $q_j = 0$ due to the label noise, then the contribution of a mislabeled sample to the gradient is reversed (i.e. gradient

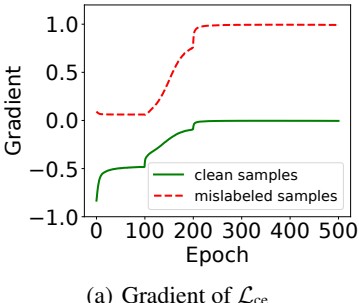
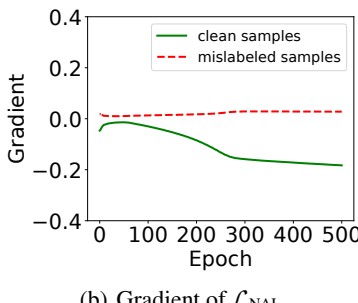

(a) Gradient of $\mathcal{L}_{\text{ce}}$            (b) Gradient of $\mathcal{L}_{\text{NAL}}$

Figure 3: The results of gradient on CIFAR-10 with 40% symmetric label noise using ResNet34. We observe that in plot (a), the gradient of clean samples dominates in early learning stage, but afterwards it vanishes (close to 0) and the gradient of mislabeled samples dominates. In plot (b), NAL effectively keeps the gradient of clean samples dominant and diminishes the gradient of mislabeled samples.

should be negative but get positive instead). The entry corresponding to the impostor class $j'$, is also reversed because $q_{j'} = 1$, causing the gradient of mislabeled samples dominates (in Figure 3(a)). Consequently, training with CE loss results in memorization of the mislabeled samples.

**Theorem 1.** *Given the noise attention loss $\mathcal{L}_{NAL}$ in Eq. (3), we rewrite the sample-wise loss $\ell_{NAL} = -\sum_{k=1}^{K} q_k \log(\tau(p_k - q_k) + q_k) - \lambda \log \tau$. Its gradient with respect to the logits $z_j$ can be derived as*

$$\frac{\partial \ell_{NAL}}{\partial z_j} = \begin{cases} \dfrac{p_y}{p_y - 1 + 1/\tau}(p_j - 1) \leq 0, & q_j = q_y = 1 \ (j \text{ is the true class for } \boldsymbol{x}), \quad (6a) \\[2ex] \dfrac{p_y}{p_y - 1 + 1/\tau}p_j \geq 0, & q_j = 0 \ (j \text{ is not the true class for } \boldsymbol{x}), \quad (6b) \end{cases}$$

*where the multiplier $\frac{p_y}{p_y - 1 + 1/\tau} \in (0, 1)$.*

Theorem 1 ensures that learning on true class persists when training with noise attention loss. In addition, compared to the gradient of $\ell_{\text{ce}}$, the gradient of $\ell_{\text{NAL}}$ is scaled by an positive multiplier term $\frac{p_y}{p_y - 1 + 1/\tau}$. Let's denote $\varphi = \frac{p_y}{p_y - 1 + 1/\tau}$. $\varphi$ is monotonically increasing on $\tau$. We have $\lim_{\tau \to 1} \varphi = 1$, and $\lim_{\tau \to 0} \varphi = 0$. For the samples with the true class $j$ in Eq. (6a), the cross-entropy gradient term $p_j - 1$ of clean samples tends to vanish after early learning stage, causing mislabeled samples to dominate the gradient. However, by multiplying $\varphi$ (note that $\varphi \to 0$ for mislabeled samples and $\varphi \to 1$ for clean samples due to attention weight distribution), it counteracts the effect of gradient dominating by mislabeled samples. For the samples that $j$ is not the true class in Eq. (6b), the gradient term $p_j$ is positive. Multiplying $\varphi < 1$ effectively dampens the magnitudes of coefficients on these mislabeled samples, thereby diminishing their effect on the gradient. Figure 3(b) empirically shows the gradient when training with $\mathcal{L}_{\text{NAL}}$. It can be observed that NAL keeps the gradient of clean samples dominant and diminishes the gradient of mislabeled samples when epoch increases, effectively preventing memorization of mislabeled samples.

## 5 Experiments

In this section, we first compare NAL with the existing methods on five widely used datasets. Then we provide several empirical results towards a better understanding of NAL. We also provide an ablation study to evaluate the influence of each component and show the results when integrated with another noise robust method SCE [23]. Finally, we discuss the scenarios when NAL encounters hard samples. All experiments are implemented in PyTorch and run in a single NIVDIA A100 GPU.

### 5.1 Comparison with Existing Methods

**Datasets and Setups.** We evaluate our approach on two benchmarks CIFAR-10 and CIFAR-100 [2] with simulated label noise, and three real-world datasets, ANIMAL-10N [15], Clothing1M [16] and WebVision [3]. Since CIFAR-10 and CIFAR-100 are initially clean, we follow [18] to inject

Table 1: Test Accuracy (%) on CIFAR-10 and CIFAR-100 with various levels of label noise injected to the training set. We compare with previous works under the same backbone ResNet34. The results are averaged over 3 trials. Results are taken from their original papers. The best results are in **bold**.

| Dataset | CIFAR-10 | | | | | CIFAR-100 | | | | |
|---|---|---|---|---|---|---|---|---|---|---|
| Noise type | symm | | | | asymm | symm | | | | asymm |
| Noise ratio | 20% | 40% | 60% | 80% | 40% | 20% | 40% | 60% | 80% | 40% |
| Cross Entropy | $86.98 \pm 0.12$ | $81.88 \pm 0.29$ | $74.14 \pm 0.56$ | $53.82 \pm 1.04$ | $80.11 \pm 1.44$ | $58.72 \pm 0.26$ | $48.20 \pm 0.65$ | $37.41 \pm 0.94$ | $18.10 \pm 0.82$ | $42.74 \pm 0.61$ |
| Forward $\hat{T}$ [18] | $87.99 \pm 0.36$ | $83.25 \pm 0.38$ | $74.96 \pm 0.65$ | $54.64 \pm 0.44$ | $83.55 \pm 0.58$ | $39.19 \pm 2.61$ | $31.05 \pm 1.44$ | $19.12 \pm 1.95$ | $8.99 \pm 0.58$ | $34.44 \pm 1.93$ |
| Bootstrap [21] | $86.23 \pm 0.23$ | $82.23 \pm 0.37$ | $75.12 \pm 0.56$ | $54.12 \pm 1.32$ | $81.21 \pm 1.47$ | $58.27 \pm 0.21$ | $47.66 \pm 0.55$ | $34.68 \pm 1.10$ | $21.64 \pm 0.97$ | $45.12 \pm 0.57$ |
| GCE [24] | $89.83 \pm 0.20$ | $87.13 \pm 0.22$ | $82.54 \pm 0.23$ | $64.07 \pm 1.38$ | $76.74 \pm 0.61$ | $66.81 \pm 0.42$ | $61.77 \pm 0.24$ | $53.16 \pm 0.78$ | $29.16 \pm 0.74$ | $47.22 \pm 1.15$ |
| Joint Opt [25] | 92.25 | 90.79 | 86.87 | 69.16 | - | 58.15 | 54.81 | 47.94 | 17.18 | - |
| NLNL [26] | 94.23 | 92.43 | 88.32 | - | 89.86 | 71.52 | 66.39 | 56.51 | - | 45.70 |
| SCE [23] | $89.83 \pm 0.20$ | $87.13 \pm 0.26$ | $82.81 \pm 0.61$ | $68.12 \pm 0.81$ | $82.51 \pm 0.45$ | $70.38 \pm 0.13$ | $62.27 \pm 0.22$ | $54.82 \pm 0.57$ | $25.91 \pm 0.44$ | $69.32 \pm 0.87$ |
| DAC [27] | 92.91 | 90.71 | 86.30 | 74.84 | - | 73.55 | 66.92 | 57.17 | 32.16 | - |
| SAT [28] | 94.14 | 92.64 | 89.23 | 78.58 | - | 75.77 | 71.38 | 62.69 | **38.72** | - |
| SELC [12] | $93.09 \pm 0.02$ | $91.18 \pm 0.06$ | $87.25 \pm 0.09$ | $74.13 \pm 0.14$ | $91.05 \pm 0.11$ | $73.63 \pm 0.07$ | $68.46 \pm 0.10$ | $59.41 \pm 0.06$ | $32.63 \pm 0.06$ | $70.82 \pm 0.09$ |
| ELR [7] | $92.12 \pm 0.35$ | $91.43 \pm 0.21$ | $88.87 \pm 0.24$ | $80.69 \pm 0.57$ | $90.35 \pm 0.38$ | $74.68 \pm 0.31$ | $68.43 \pm 0.42$ | $60.05 \pm 0.78$ | $30.27 \pm 0.86$ | $73.73 \pm 0.34$ |
| NAL (ours) | **94.37 ± 0.04** | **93.49 ± 0.07** | **90.56 ± 0.07** | **80.98 ± 0.27** | **92.09 ± 0.12** | **77.79 ± 0.28** | **74.65 ± 0.09** | **68.48 ± 0.16** | $36.77 \pm 0.71$ | **74.73 ± 0.12** |

Table 2: Test accuracy (%) on CIFAR under different types of PMD noise [20]. All methods use a PreActResNet34 architecture.

| Dataset | Noise | Cross Entropy | SCE [23] | LRT [29] | PLC [20] | SELC [12] | NAL (ours) |
|---|---|---|---|---|---|---|---|
| CIFAR-10 | Type-I (35%) | $78.11 \pm 0.74$ | $79.76 \pm 0.72$ | $80.98 \pm 0.80$ | $82.80 \pm 0.27$ | $86.97 \pm 0.15$ | **88.81 ± 0.13** |
| | Type-II (35%) | $76.65 \pm 0.57$ | $77.92 \pm 0.89$ | $80.74 \pm 0.25$ | $81.54 \pm 0.47$ | $87.06 \pm 0.20$ | **87.66 ± 0.23** |
| | Type-III (35%) | $76.89 \pm 0.79$ | $78.81 \pm 0.29$ | $81.08 \pm 0.35$ | $81.50 \pm 0.50$ | $87.31 \pm 0.18$ | **88.57 ± 0.16** |
| CIFAR-100 | Type-I (35%) | $57.68 \pm 0.29$ | $55.20 \pm 0.33$ | $56.74 \pm 0.34$ | $60.01 \pm 0.43$ | $65.72 \pm 0.17$ | **66.55 ± 0.16** |
| | Type-II (35%) | $57.83 \pm 0.25$ | $56.10 \pm 0.73$ | $57.25 \pm 0.68$ | $63.68 \pm 0.29$ | $66.79 \pm 0.18$ | **67.15 ± 0.11** |
| | Type-III (35%) | $56.07 \pm 0.79$ | $56.04 \pm 0.74$ | $56.57 \pm 0.30$ | $63.68 \pm 0.29$ | $66.41 \pm 0.17$ | **66.59 ± 0.40** |

Table 3: The accuracy (%) results on ANIMAL-10N. All methods use an VGG-19 architecture. Results of other methods are taken from original papers.

| Cross Entropy | Nested [30] | SELFIE [15] | PLC [20] | SELC [12] | Nested + Co-teaching [30] | NAL (ours) |
|---|---|---|---|---|---|---|
| $79.40 \pm 0.14$ | $81.30 \pm 0.60$ | $81.80 \pm 0.09$ | $83.40 \pm 0.43$ | $83.73 \pm 0.06$ | $84.10 \pm 0.10$ | **84.18 ± 0.19** |

Table 4: The accuracy (%) results on Clothing1M. All compared methods use the ResNet-50 pretrained on ImageNet. The marker ‡ denotes the model is trained from scratch.

| Cross Entropy | SCE [23] | DMI [31] | ODNL [32] | ELR [7] | FINE [33] | Nested [30] | HOC [34] | NAL‡ (ours) |
|---|---|---|---|---|---|---|---|---|
| 69.21 | 71.02 | 72.46 | 72.47 | 72.87 | 72.91 | 73.10 | 73.39 | **73.58** |

Table 5: The accuracy (%) results on (mini) Webvision. Results of other methods are taken from their original papers. All methods use an InceptionResNetV2 architecture.

| | | Co-teaching [8] | Iterative-CV [35] | CRUST [36] | ODD [37] | NCT [38] | ELR [7] | NAL (ours) |
|---|---|---|---|---|---|---|---|---|
| WebVision | top1 | 63.58 | 65.24 | 72.40 | 74.60 | 75.16 | 76.26 | **77.41** |
| | top5 | 85.20 | 85.34 | 89.56 | 90.60 | 90.77 | 91.26 | **92.25** |
| ILSVRC12 | top1 | 61.48 | 61.60 | 67.36 | 57.80 | 71.73 | 68.71 | **74.09** |
| | top5 | 84.70 | 84.98 | 87.84 | 86.30 | 91.61 | 87.84 | **92.09** |

class-conditional label noise: symmetric and asymmetric label noises. Symmetric noise is generated by uniformly flipping the label to one of the other class label. Asymmetric noise is a simulation of fine-grained classification, where the label flipping only occurs within very similar classes (e.g. dog ↔ cat). Following PLC [20], we also evaluate NAL on Polynomial Margin Diminishing (PMD) noise, an instance-dependent noise that allows arbitrary noise strength in a wide buffer near the decision

boundary. ANIMAL-10N contains human-labeled online images for 10 animals with confusing appearance. Its estimated noise rate is 8%. Clothing1M consists of 1 million images collected from online shopping websites with labels generated from surrounding texts. Its estimated noise rate is 38.5%. WebVision contains 2.4 million images crawled from the web using the 1000 concepts in ImageNet ILSVRC12. Its estimated noise rate is 20%. For CIFAR with class-conditional noise, we use a ResNet-34 [39] and train it using SGD with a batch size of 64. For CIFAR with instance-dependent noise, we use a PreActResNet34 [14] and train it using SGD with a batch size of 128. For ANIMAL-10N, we use VGG-19 [40] with batch normalization and train it using SGD with a batch size of 128. For Clothing1M, we train a ResNet-50 [39] using SGD with a batch size of 64. For Webvision, we train a InceptionResNetV2 [41] using SGD with a batch size of 32. More training details can be found in Appendix B.

Table 1 shows the results on CIFAR-10 and CIFAR-100 with different levels of symmetric and asymmetric label noise. We compare NAL to the best performing approaches that only modify the training loss. All of these methods use the same backbone (ResNet34). NAL shows substantial improvements over other methods and obtains the highest accuracy in most cases.

Table 2 shows the results on three types of instance-dependent (PMD) noise from PLC [20]. We observe that the proposed method outperforms baselines across different noise settings.

Table 3 shows the results on ANIMAL-10N. NAL achieves state-of-the-art performance, even better than Nested which uses Co-teaching to boost performance.

Table 4 shows the results on the Clothing1M. All compared methods are trained using ResNet50 pretrained on ImageNet. However, as NAL needs to learn the parameters $W$ and $b$ in the attention branch, we directly train our model from scratch. As we can observe from Table 4, NAL still slightly outperforms other methods even without using an ImageNet pretrained model.

Table 5 shows the results on the (mini) WebVision following [35]. All the compared methods are evaluated on WebVision and ImageNet ILSVRC12 validation sets. We observe that NAL consistently outperforms state-of-the-art methods on both validation sets in terms of top1 and top5 accuracy.

Note that we do not compare with some state-of-the-art methods like RoCL [42] and DivideMix [10] as baselines, because these methods are aggregations of multiple techniques (e.g. mixup [43], multiple networks [8], and complex augmentation [44]), while this paper only focuses on one, so the comparison seems unfair.

## 5.2 Empirical Analysis of NAL

**Memorization Procedure of NAL.** Figure 1(c) and 1(d) show the memorization procedure of NAL on clean and mislabeled samples, respectively. Compared to CE loss in Figure 1(a) and 1(b), NAL effectively prevents memorization of mislabeled samples, allowing the model to continue learning on the clean samples to attain high classification accuracy.

**Gradient of $\mathcal{L}_{ce}$ and $\mathcal{L}_{NAL}$.** Figure 3(a) and 3(b) show the gradients of CE loss and noise attention loss on CIFAR-10 with 40% symmetric noise using ResNet34 [39] respectively. In comparison to CE loss, the proposed method keeps the gradient of clean samples dominant and dampens the gradient of mislabeled samples, resulting in the whole gradient being guided by clean samples.

**Attention Weights.** When trained on noisy labels, we have observed that the mislabeled samples have smaller weights than the clean samples in Figure 2. Here, we report the average weights of samples in Figure 4(a). The $(i, j)$-th block represents the average weights of samples with clean label $i$ and noisy label $j$. We observe that the attention weights on the diagonal blocks are higher than those on non-diagonal blocks, which indicates NAL does properly down-weight the mislabeled samples.

**Quality of Estimated Targets.** We evaluate the quality of new targets $t$ by correction accuracy defined via $\frac{1}{N} \sum_i^N \mathbb{1}\{\arg\max_j y_j^{[i]} = \arg\max_j t_j^{[i]}\}$, where $y^{[i]}$ is the true label of input $x^{[i]}$. Figure 4(d) shows the correction accuracy vs. epochs on CIFAR-10 with different levels of label noise. The target estimation in Section 3.3 steadily improves the quality of new targets. Figure 4(b) and Figure 4(c) shows the confusion matrix of noisy labels and corrected labels (i.e. the hard version of new targets) w.r.t the clean labels on CIFAR-10 with 60% symmetric noise respectively. We observe that target estimation corrects the noisy labels impressively well for all classes.

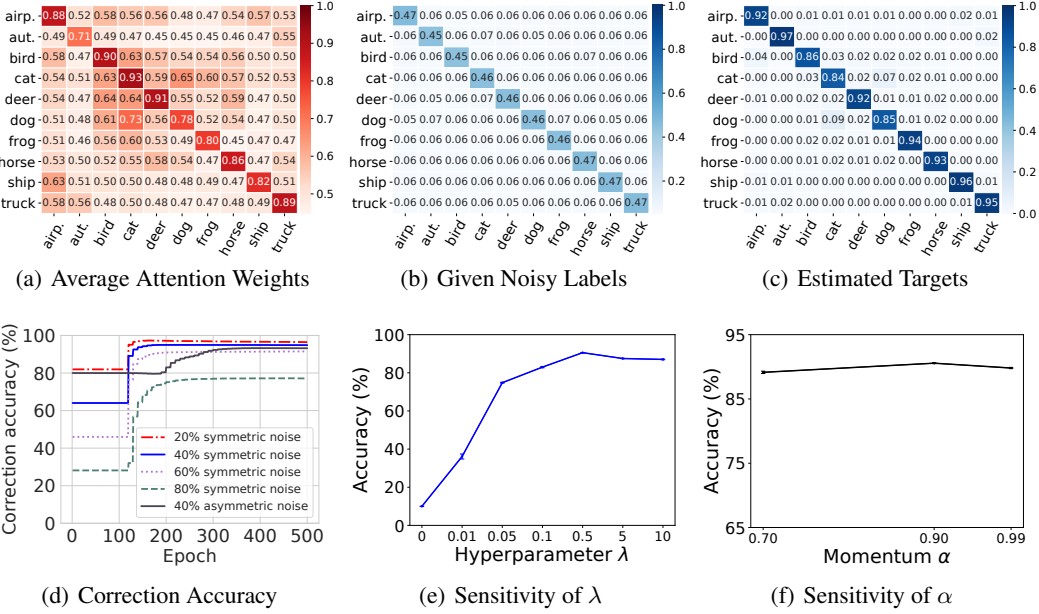

(a) Average Attention Weights  (b) Given Noisy Labels  (c) Estimated Targets

(d) Correction Accuracy  (e) Sensitivity of $\lambda$  (f) Sensitivity of $\alpha$

Figure 4: (a). Average attention weights of noisy labels w.r.t clean labels on CIFAR-10 with 60% symmetric noise. (b). Confusion matrix of noisy labels w.r.t true labels on CIFAR-10 with 60% symmetric noise. (c). Confusion matrix of corrected labels w.r.t true labels on CIFAR-10 with 60% symmetric noise. (d). Label correction accuracy vs. epochs on CIFAR-10 with different levels of noise. Plots (e) and (f) show the sensitivity of $\lambda$ and $\alpha$ on CIFAR-10 with 60% symmetric noise.

**Sensitivity of Hyperparameters $\lambda$ and $\alpha$.** NAL contains two hyperparameters: $\lambda$ is the coefficient for boost term $\mathcal{L}_{\mathrm{b}}$, $\alpha$ is the momentum in target estimation. Figure 4(e) and 4(f) show their sensitivity to the performance. We observe that the sensitivity to $\alpha$ is quite mild as long as it is set close to 1. $\lambda$ need to be tuned according to the complexity of dataset. It cannot be set to be very large or very small, resulting in neglecting attention term or falling into the 'lazy' learning circumstance. In our experiments, we fixed $\alpha = 0.9$ for all datasets. For CIFAR-10 and ANIMAL-10N, we set $\lambda = 0.5$. For CIFAR-100, we set $\lambda = 10$. For Webvision and Clothing1M, we set $\lambda = 50$.

## 5.3  Ablation Study

Table 6 reports the influence of the two components in NAL: target estimation and attention branch. Removing the target estimation leads to a significant performance drop. This suggests that using target estimation is crucial as it generates more clean samples for the model to learn. To validate the effect of the attention branch, we conduct another way to calculate the attention weights: using the confidence score (highest probability in output) as the weight, i.e., $\max_j \boldsymbol{p}_j, j \in [1, K]$. We observe that the model does not converge in hard cases. We conjecture that using the confidence from model output does interfere with the original prediction branch, while adding the attention branch to get attention weight effectively solve this problem. We also explore the possibility of integrating NAL with other methods. Here we adopt the idea from symmetric cross entropy (SCE) [23], which adds the reverse term to achieve noise robustness. As shown in Table 6, adding RNAL affects the performance under CIFAR-10, but enjoys performance boost under CIFAR-100 without extra cost.

## 5.4  Discussion on Hard Samples

We assume the training set consists of mislabeled samples, easy (clean) samples and hard (clean) samples. In this work, we do not design a specific technique to distinguish hard samples but the proposed NAL can handle hard samples as well. In the early learning phase, both mislabeled samples and hard samples have lower weights compared to the easy samples, which forces the model to learn only from easy samples due to gradient scaling (Theorem 1). After the model has learned from easy samples for a while, the model then gradually learns from hard samples rather than the mislabeled

Table 6: Ablation study results in terms of test accuracy (%) on CIFAR-10 and CIFAR-100. ⊖ means the model fails to converge. + RNAL means adding the reverse term of noise attention loss.

| Dataset | CIFAR-10 | | | CIFAR-100 | | |
|---|---|---|---|---|---|---|
| Noise type | symm | | asymm | symm | | asymm |
| Noise ratio | 40% | 80% | 40% | 40% | 80% | 40% |
| NAL | **93.49 ± 0.07** | **80.98 ± 0.27** | **92.09 ± 0.12** | 74.65 ± 0.09 | 36.77 ± 0.71 | 74.73 ± 0.12 |
| NAL w/o target estimation | 89.47 ± 0.50 | 76.91 ± 0.22 | 88.23 ± 0.22 | 69.91 ± 0.21 | 31.33 ± 0.38 | 55.68 ± 0.17 |
| NAL w/o attention branch | 90.94 ± 0.28 | ⊖ | 91.55 ± 0.07 | ⊖ | ⊖ | ⊖ |
| NAL + RNAL | 92.92 ± 0.29 | 80.20 ± 0.42 | 88.17 ± 0.63 | **75.38 ± 0.08** | **38.24 ± 0.55** | **74.89 ± 0.20** |

samples. This is because the hard samples are more connected to the easy samples than the mislabeled samples, as the former two share some common features [45, 46]. Therefore, the proposed method can continuously learn from clean samples (both easy and hard) and prevent the memorization of mislabeled samples. In Figure 2, we can observe less overlap between clean and mislabeled samples. We also observe that some clean samples are given low weights, and some mislabeled samples are given high weights. The clean samples given low weights are actually hard clean samples. There are also mislabeled samples given high weights. These mislabeled samples are hard samples with "proper" wrong labels. For example, given a hard sample: a cat image that looks like a dog due to blurry resolution (e.g. $32 \times 32$). When generating simulated label noise, it is likely to assign a dog label for this sample. Compared to regular mislabeled samples, this kind of mislabeled samples can be learned easily by the classifier, resulting in having high weights.

## 6 Related Work

We briefly discuss the existing noise-robust methods that do not require a set of clean training data (as opposed to [16, 47, 48]). **Loss Correction.** Many approaches focus on correcting the loss function explicitly by estimating the noise transition matrix [49, 18, 50]. **Robust Loss Functions.** These studies develop loss functions that are robust to label noise, including $\mathcal{L}_{\text{DMI}}$ [31], MAE [51], GCE [24], SCE [23], NCE [52], TCE [53] and GJS [54]. Above two categories of methods do not utilize the early learning phenomenon. **Sample Selection** During the early learning stage, the samples with smaller loss values are more likely to be the clean samples. Based on this observation, MentorNet [55] pre-trains a mentor network for selecting small-loss samples to guide the training of the student network. Co-teaching related methods [8, 56–58] maintain two networks, and each network is trained on the small-loss samples selected by its peer network. **Label Correction** Joint Opt [25] and PENCIL [59] replace the noisy labels with soft (i.e. model probability) or hard (i.e to one-hot vector) pseudo-labels. [9] weigh the clean and mislabeled samples by fitting a two-component Beta mixture model to loss values, and corrects the labels via mixup combination. Similarly, DivideMix [10] trains two networks to separate the clean and mislabeled samples via a two-component Gaussian mixture model, and further uses MixMatch [60] to enhance the performance. **Regularization** [61] prove the gradient descent with early stopping is an effective regularization to achieve robustness to label noise. [62] *explicitly* add the regularizer based on neural tangent kernel [63] to limit the distance between the model parameters to initialization. ELR [7] estimates the target by temporal ensembling [13] and adds a regularization term to cross-entropy loss to avoid memorization. Other techniques, such as mixup augmentation [43], label smoothing [41] and weight averaging [64], can effectively improve the performance under label noise.

## 7 Conclusion

In this work, we propose NAL for learning with noisy labels. Our method leverages an attention branch and a noise attention loss to learn the attention weights for distinguishing the mislabeled samples from clean samples. NAL can effectively diminish the gradient of mislabeled samples, mitigating the effect of noisy labels. We also provide extensive empirical analyses and evaluate its effectiveness across multiple datasets with different types and ratios of label noises.

There are still multiple open problems for future research. Currently, label noise has been extensively studied in image classification task. Other research areas, such as graph learning and federated learning, remain to be explored. On the methodological front, we hope that our work will trigger interest in designing novel network architectures that inherently provide robustness to label noise.

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
