# Appendix

**Yangdi Lu**
Department of Computing and Software
McMaster University
`luy100@mcmaster.ca`

**Yang Bo**
Department of Computing and Software
McMaster University
`boy2@mcmaster.ca`

**Wenbo He**
Department of Computing and Software
McMaster University
`hew11@mcmaster.ca`

## A   Missing Proofs

### A.1   Gradient Derivation of $\ell_{\text{NAL}}$

The sample-wise $\ell_{\text{NAL}}$ can be rewrite as:

$$\ell_{\text{NAL}} = \ell_{\text{na}} + \lambda \ell_{\text{b}} = -\sum_{k=1}^{K} q_k \log(\tau(p_k - q_k) + q_k) - \lambda \log \tau. \tag{1}$$

As $\tau$ is learned from attention branch. The derivation of the $\ell_{\text{NAL}}$ with respect to the logits is as follows:

$$\frac{\partial \ell_{\text{NAL}}}{\partial z_j} = \frac{\partial \ell_{\text{na}}}{\partial z_j} = -\sum_{k=1}^{K} \frac{\tau q_k}{\tau(p_k - q_k) + q_k} \frac{\partial p_k}{\partial z_j}. \tag{2}$$

Since $p_k = \texttt{softmax}(\boldsymbol{z}) = \frac{e^{z_k}}{\sum_{j=1}^{K} e^{z_j}}$, we have

$$\frac{\partial p_k}{\partial z_j} = \frac{\partial \left( \frac{e^{z_k}}{\sum_{j=1}^{K} e^{z_j}} \right)}{\partial z_j} = \frac{\frac{\partial e^{z_k}}{\partial z_j}(\sum_{j=1}^{K} e^{z_j}) - e^{z_k} \frac{\partial \left( \sum_{j=1}^{K} e^{z_j} \right)}{\partial z_j}}{(\sum_{j=1}^{K} e^{z_j})^2}. \tag{3}$$

In the case of $k = j$ :

$$\begin{aligned}
\frac{\partial p_k}{\partial z_j} &= \frac{\frac{\partial e^{z_k}}{\partial z_k}(\sum_{k=1}^{K} e^{z_k}) - e^{z_k} \frac{\partial \left( \sum_{k=1}^{K} e^{z_k} \right)}{\partial z_k}}{(\sum_{k=1}^{K} e^{z_k})^2} = \frac{e^{z_k}(\sum_{k=1}^{K} e^{z_k}) - e^{z_k} \cdot e^{z_k}}{(\sum_{k=1}^{K} e^{z_k})^2} \\
&= \frac{e^{z_k}}{\sum_{k=1}^{K} e^{z_k}} - \left( \frac{e^{z_k}}{\sum_{k=1}^{K} e^{z_k}} \right)^2 = p_k - p_k^2.
\end{aligned} \tag{4}$$

In the case of $k \neq j$ :

$$\frac{\partial p_k}{\partial z_j} = \frac{0 \cdot (\sum_{j=1}^{K} e^{z_j}) - e^{z_k} \cdot e^{z_j}}{(\sum_{j=1}^{K} e^{z_j})^2} = -\frac{e^{z_k}}{\sum_{j=1}^{K} e^{z_j}} \frac{e^{z_j}}{\sum_{j=1}^{K} e^{z_j}} = -p_k p_j. \tag{5}$$

36th Conference on Neural Information Processing Systems (NeurIPS 2022).

Combining Eq. (4) and (5) into Eq. (2), we obtain:

$$
\begin{aligned}
\frac{\partial \ell_{\text{NAL}}}{\partial z_j} &= -\sum_{k=1}^{K} \frac{\tau q_k}{\tau(p_k - q_k) + q_k} \frac{\partial p_k}{\partial z_j} \\
&= -\frac{\tau q_j}{\tau(p_j - q_j) + q_j} \frac{\partial p_j}{\partial z_j} - \sum_{k \neq j}^{K} \frac{\tau q_k}{\tau(p_k - q_k) + q_k} \frac{\partial p_k}{\partial z_j} \\
&= -\frac{\tau q_j}{\tau(p_j - q_j) + q_j}(p_j - p_j^2) - \sum_{k \neq j}^{K} \frac{\tau q_k}{\tau(p_k - q_k) + q_k}(-p_k p_j) \\
&= -\frac{\tau q_j p_j}{\tau(p_j - q_j) + q_j} + p_j \sum_{k=1}^{K} \frac{\tau q_k p_k}{\tau(p_k - q_k) + q_k}.
\end{aligned}
\tag{6}
$$

Therefore, if $q_j = q_y = 1$, then

$$
\frac{\partial \ell_{\text{NAL}}}{\partial z_j} = -\frac{\tau p_j}{\tau p_j - \tau + 1} + p_j \frac{\tau q_j p_j}{\tau(p_j - 1) + 1} = (p_j - 1)\frac{\tau p_j}{\tau p_j - \tau + 1} = (p_j - 1)\frac{p_j}{p_j - 1 + 1/\tau}.
\tag{7}
$$

If $q_j = 0$, then

$$
\frac{\partial \ell_{\text{NAL}}}{\partial z_j} = p_j \frac{\tau q_y p_y}{\tau(p_y - q_y) + q_y} = p_j \frac{p_y}{p_y - 1 + 1/\tau}.
\tag{8}
$$

## A.2 Formal Proofs for Theorem 1

**Theorem 1.** *Given the noise attention loss $\mathcal{L}_{NAL}$, we rewrite the sample-wise loss $\ell_{NAL} = -\sum_{k=1}^{K} q_k \log(\tau(p_k - q_k) + q_k) - \lambda \log \tau$. Its gradient with respect to the logits $z_j$ can be derived as*

$$
\frac{\partial \ell_{NAL}}{\partial z_j} =
\begin{cases}
\dfrac{p_y}{p_y - 1 + 1/\tau}(p_j - 1) \leq 0, & q_j = q_y = 1 \ (j \text{ is the true class for } \boldsymbol{x}) & \text{(9a)} \\[3mm]
\dfrac{p_y}{p_y - 1 + 1/\tau}p_j \geq 0, & q_j = 0 \ (j \text{ is not the true class for } \boldsymbol{x}) & \text{(9b)}
\end{cases}
$$

*where the multiplier $\frac{p_y}{p_y - 1 + 1/\tau} \in (0, 1)$.*

*Proof.* From the Appendix A.1, we obtain the gradient of the sample-wise $\ell_{\text{NAL}}$ with respect to the logits $z_j$ is

$$
\frac{\partial \ell_{\text{NAL}}}{\partial z_j} = -\sum_{k=1}^{K} \frac{\tau q_k}{\tau(p_k - q_k) + q_k} \frac{\partial p_k}{\partial z_j}
\tag{10}
$$

where $\frac{\partial p_k}{\partial z_j}$ can be further derived base on whether $k = j$ by follows:

$$
\frac{\partial p_k}{\partial z_j} =
\begin{cases}
p_k - p_k^2 & k = j \\
-p_j p_k & k \neq j
\end{cases}
\tag{11}
$$

According to Eq. (10) and (11), the gradient of $\ell_{\text{NAL}}$ can be derived as:

$$
\frac{\partial \ell_{\text{NAL}}}{\partial z_j} =
\begin{cases}
\frac{p_j}{p_j - 1 + 1/\tau}(p_j - 1) = \frac{p_y}{p_y - 1 + 1/\tau}(p_j - 1), & q_j = q_y = 1 \\[3mm]
\frac{p_y}{p_y - 1 + 1/\tau}p_j, & q_j = 0
\end{cases}
\tag{12}
$$

We denote $\varphi = \frac{p_y}{p_y - 1 + 1/\tau}$. Since $p_j \leq 1$, we have $p_j - 1 \leq 0$. As $\tau \in (0, 1)$, the term $\frac{p_y}{p_y - 1 + 1/\tau} \in (0, 1)$, we have $(p_j - 1)\frac{p_y}{p_y - 1 + 1/\tau} \leq 0$ and $p_j \frac{p_y}{p_y - 1 + 1/\tau} \geq 0$. $\qquad \square$

Table 1: Description of the datasets used in the experiments.

| Dataset | # of train | # of val | # of test | # of classes | input size | Noise rate (%) |
|---------|-----------|----------|-----------|--------------|------------|----------------|
| Datasets with clean annotation | | | | | | |
| CIFAR-10 | 50K | - | 10K | 10 | $32 \times 32$ | $\approx 0.0$ |
| CIFAR-100 | 50K | - | 10K | 100 | $32 \times 32$ | $\approx 0.0$ |
| Datasets with real world noisy annotation | | | | | | |
| ANIMAL-10N | 50K | - | 5K | 10 | $64 \times 64$ | $\approx 8$ |
| Clothing1M | 1M | 14K | 10K | 14 | $224 \times 224$ | $\approx 38.5$ |
| Webvision 1.0 | 66K | - | 2.5K | 50 | $256 \times 256$ | $\approx 20.0$ |

## B Detail Description of Experiments

Source code for the experiments is available in the zip file. All experiments are implemented in PyTorch and run in a single Nvidia A100 GPU. For CIFAR-10 and CIFAR-100, we do not perform early stopping since we don't assume the presence of clean validation data. All test accuracy are recorded from the last epoch of training. For Clothing1M, it provides 50k, 14k, 10k refined clean data for training, validation and testing respectively. Note that we do not use the 50k clean data for fair comparison with existing methods. Similar to the compared methods [1, 2], we report the test accuracy when the performance on validation set is optimal. All tables of CIFAR-10/CIFAR-100 report the mean and standard deviation from 3 trails with different random seeds for different simulated noise.

### B.1 Dataset Description and Preprocessing

The information of datasets are described in Table 1. CIFAR-10 and CIFAR-100 are clean datasets, we describe the label noise injection in Appendix B.2. ANIMAL-10N contains human-labeled online images for 10 animals with confusing appearance. Its estimated noise rate is 8%. Clothing1M consists of 1 million training images from 14 categories collected from online shopping websites with noisy labels generated from surrounding texts. Its noise level is estimated as 38.5% [3]. Following [4, 5], we use the mini WebVision dataset which contains the top 50 classes from the Google image subset of WebVision, which results in approximate 66 thousand images. The noise level of WebVision is estimated at 20% [6].

As for data preprocessing, we apply normalization and regular data augmentation (i.e. random crop and horizontal flip) on the training sets of all datasets. The cropping size is consistent with existing works [1, 7]. Specifically, 32 for CIFAR-10 and CIFAR-100, $224 \times 224$ for Clothing 1M (after resizing to $256 \times 256$), and $227 \times 227$ for Webvision.

### B.2 Simulated Label Noise Injection

Since the CIFAR-10 and CIFAR-100 are initially clean, we follow [8, 9] for symmetric and asymmetric label noise injection. Specifically, symmetric label noise is generated by randomly flipping a certain fraction of the labels in the training set following a uniform distribution. Asymmetric label noise is simulated by flipping their class to another certain class according to the mislabel confusions in the real world. For CIFAR-10, the asymmetric noisy labels are generated by mapping *truck* $\rightarrow$ *automobile*, *bird* $\rightarrow$ *airplane*, *deer* $\rightarrow$ *horse* and *cat* $\leftrightarrow$ *dog*. For CIFAR-100, the noise flips each class into the next, circularly within super-classes.

### B.3 Training Procedure

**CIFAR-10/CIFAR-100**: We use a ResNet-34 and train it using SGD with a momentum of 0.9, a weight decay of 0.001, and a batch size of 64. The network is trained for 500 epochs for both CIFAR-10 and CIFAR-100. We use the cosine annealing learning rate [10] where the maximum number of epoch for each period is 10, the maximum and minimum learning rate is set to 0.02 and 0.001 respectively. Note that the reason that we train the model 500 epochs in total is not because of the

slow convergence (Our method actually converges around 250 epochs, shown in Section E). Instead, it is to fully evaluate whether the proposed method will overfit mislabeled samples, which avoids the interference caused by early stopping [11] (i.e. the model may not start overfitting mislabeled samples when the number of training epochs is small, especially when learning rate scheduler is cosine annealing [10]).

**Clothing1M**: Following [12, 2], we use a ResNet-50 without pretrained parameters. We train the model with batch size 64. The optimization is done using SGD with a momentum 0.9, and weight decay 0.001. We use the same cosine annealing learning rate as CIFAR-10 except the minimum learning rate is set to 0.0001 and total epoch is 250. For each epoch, we sample 2000 mini-batches from the training data ensuring that the classes of the noisy labels are balanced.

**Webvision**: Following [7, 1], we use an InceptionResNetV2 as the backbone architecture. All other optimization details are the same as for CIFAR-10, except for the weight decay (0.0005) and the batch size (32).

### B.4 Hyperparameters Selection and Sensitivity

We perform hyperparameter tuning via grid search: $\lambda = [0.01, 0.05, 0.1, 0.5, 5, 10, 50]$ and $\alpha = [0.7, 0.9, 0.99]$ using a noisy validation set sampled from the noisy training set, which is similar to [1]. In our experiments, we set $\alpha = 0.9$ for all datasets. For CIFAR-10, we set $\lambda = 0.5$. For CIFAR-100, we set $\lambda = 10$. For ANIMAL-10N, we set $\lambda = 0.5$. For Webvision and Clothing1M, we set $\lambda = 50$.

## C   More Results on Training Clothing1M from Scratch

Existing methods use the ResNet-50 pretrained on ImageNet. Here we report the results when existing methods train the model from scratch in Table 2. We run the official code of SCE, SELC, DivideMix and ELR+. SCE [2] is a noise robust loss function. SELC [13] is a label correction method. These two methods only focus on modifying the loss function, thus no extra GPU space is required. In contrast, DivideMix and ELR+ (i.e. an improved version of ELR) are complex method that use multiple techniques to boost their performance. Both of them use the two networks and mixup augmentations. Other technique, such as weight average is also applied in ELR+. Therefore, more GPU space is required for DivideMix and ELR+. As we can observe in Table 2, the proposed method NAL still outperforms these methods when training the ResNet-50 from scratch.

Table 2: The accuracy (%) results on Clothing1M. The marker † denotes the methods use ResNet-50 pretrained on ImageNet. The marker ‡ denotes the model is trained from scratch.

| Method | Batch Size | Required GPU Memory | Accuracy |
|---|---|---|---|
| SCE† [2] | 64 | 7.39 GB | 71.02 |
| SCE‡ [2] | 64 | 7.39 GB | 69.40 |
| SELC† [13] | 64 | 7.39 GB | 74.01 |
| SELC‡ [13] | 64 | 7.39 GB | 72.02 |
| ELR† [1] | 64 | - | 72.87 |
| ELR+† [1] | 64 | 21.70 GB | 74.81 |
| ELR+‡ [1] | 64 | 21.70 GB | 72.80 |
| DivideMix† [7] | 32 | 19.03 GB | 74.76 |
| DivideMix‡ [7] | 32 | 19.03 GB | 70.29 |
| NAL‡ (ours) | 64 | 7.46 GB | 73.58 |

## D   More results of Estimated Targets

We report the confusion matrix for other levels of label noise (symmetric 40%, 80% and asymmetric 40%) on CIFAR-10. Figure 1, Figure 3 and Figure 5 display the confusion matrix of noisy labels w.r.t.

the clean labels on CIFAR-10 with 40% symmetric, 80% symmetric and 40% asymmetric label noise respectively. Figure 2, Figure 4 and Figure 6 display the confusion matrix of corrected labels w.r.t. the clean labels on CIFAR-10 with 40% symmetric, 80% symmetric and 40% asymmetric label noise after using the proposed method, respectively. As we can observe, the corrected labels (estimated targets) yield better quality than the original noisy labels, even under the extreme label noise (e.g. 80% symmetric noise).

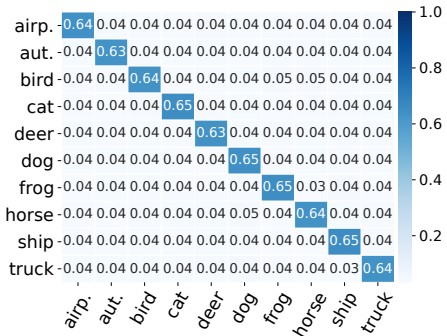

Figure 1: Confusion matrix of noisy labels w.r.t clean labels on CIFAR-10 with 40% symmetric label noise.

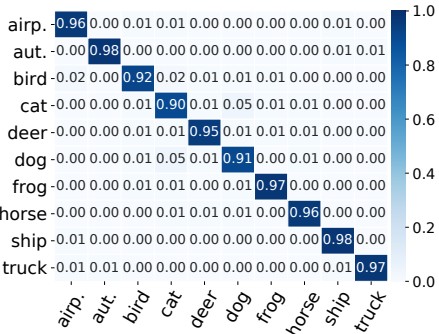

Figure 2: Confusion matrix of corrected labels w.r.t clean labels on CIFAR-10 with 40% symmetric label noise.

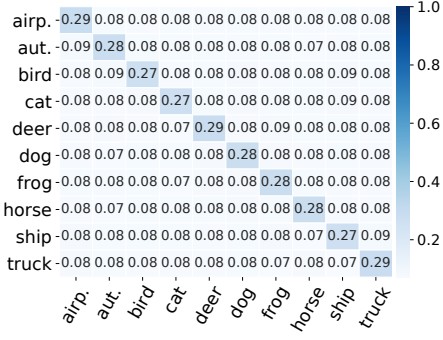

Figure 3: Confusion matrix of noisy labels w.r.t clean labels on CIFAR-10 with 80% symmetric label noise.

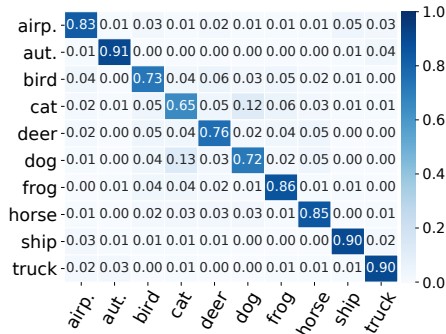

Figure 4: Confusion matrix of corrected labels w.r.t clean labels on CIFAR-10 with 80% symmetric label noise.

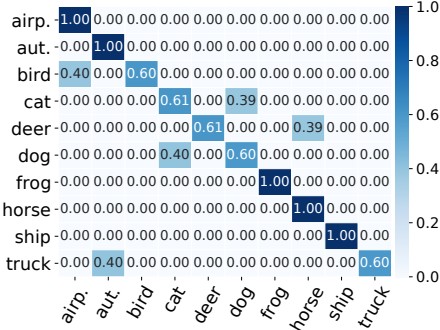

Figure 5: Confusion matrix of noisy labels w.r.t clean labels on CIFAR-10 with 40% asymmetric label noise.

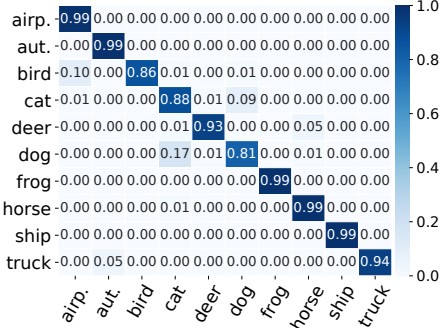

Figure 6: Confusion matrix of corrected labels w.r.t clean labels on CIFAR-10 with 40% asymmetric label noise.

# E    Accuracy Curves versus Epochs

In this section, we show the accuracy curves during training to show the noise robustness of NAL. As we can observe in Figure 7, CE fits the whole noisy labels eventually, while NAL only fits around 60% training samples. In Figure 8, the test accuracy of CE decreases due to memorization of noisy labels, while NAL won't. We also observe the similar results on CIFAR-10.

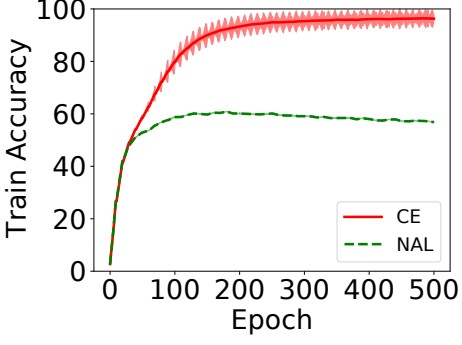

Figure 7: Train accuracy vs. epochs on CIFAR-100 with 40% symmetric label noise.

Figure 8: Test accuracy vs. epochs on CIFAR-100 with 40% symmetric label noise.

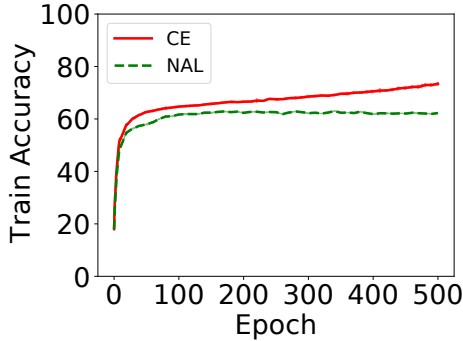

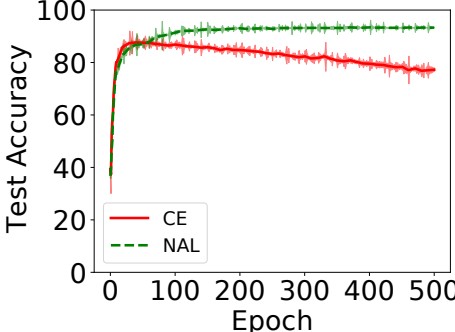

Figure 9: Train accuracy vs. epochs on CIFAR-10 with 40% symmetric label noise.

Figure 10: Test accuracy vs. epochs on CIFAR-10 with 40% symmetric label noise.

# F    Movement of weight $\tau$ Distribution

We monitor the weight $\tau$ distribution during training on CIFAR-10 with 40% symmetric noise using NAL. Figure 11 to Figure 13 show the results. We summarize the weight $\tau$ distribution into three stages. As can be observed, in the stage 1, clean samples and mislabeled samples have the similar $\tau$. By continuing to train the model with NAL, the weights for clean samples and mislabeled samples start to separate. Until stage 3, the weight of clean samples is increased towards 1.0, even for most hard samples (since the weight distribution has only a little overlap between clean and mislabeled samples).