# OpenReview forum: "Noise Attention Learning: Enhancing Noise Robustness by Gradient Scaling"
_NeurIPS.cc/2022/Conference — NeurIPS 2022 Accept_

### Official Review · Reviewer_BcSR · 2022-07-07

**Rating:** 6
**Confidence:** 4
**Soundness:** 3 good
**Presentation:** 3 good
**Contribution:** 3 good

**Summary:**

The authors proposed a simple method that trains an attention layer to generate weights for different samples. By using neural networks’ memorization effects, the weights can be learned during the training process. This method simplifies the training process, which does not need to extract confident examples. The authors also give theoretical analyses of their method. The proposed method experimentally shows large improvements compared with baselines.

**Questions:**

1. Although the empirical results are good, could the authors simply explain how this proposed attention branch distinguishes mislabelled samples and some hard examples that are not learned in the early learning phase?
2. For CIFAR-10 with 60% in figure 2, I found some noise is given with high weights, and some clean data is given low weights. Can you explain it?
3. I found the performance is lower than some state-of-the-art methods, and the authors claim that it does not employ some techniques like MixUp, or two networks. Does the proposed method combine well with these widely used techniques in learning with label noise, for example, MixUp?


**Limitations:**

1. From my perspective, the word “attention” looks a little ambiguous. It more seems like a confidence score for each sample.
2. The proposed method builds on semi-technique ensemble predictions. So, I think the authors should compare it with state-of-the-art methods that also use semi-technique e.g., DivideMix, FINE[1], or PES [2], which makes it easy to compare.
3. The authors only conduct symmetric label noise and asymmetric noise. Some commonly used settings like Pairflip 45% and Instance-dependent label noise [3, 4] are missing.
4. The hyperparameter lambda is sensitive and has a large range for different datasets.

[1] FINE Samples for Learning with Noisy Labels
[2] Understanding and Improving Early Stopping for Learning with Noisy Labels
[3] LEARNING WITH FEATURE-DEPENDENT LABEL NOISE: A PROGRESSIVE APPROACH
[4] Part-dependent Label Noise: Towards Instance-dependent Label Noise Xiaobo


**Strengths And Weaknesses:**

1. Training an attention layer, directly outputting a score, is novel.
2. The proposed method is simple and empirical works well.
3. This paper is well organized and easy to understand.

---

> ### Author Response · Authors · 2022-08-01
> **Response to Reviewer BcSR**
>
> Thank you for an insightful review and questions. Please see below for answers to your questions.
>
> **Q1. About hard samples.**
> **A1.** We assume the training set consists of mislabeled samples, easy (clean) samples and hard (clean) samples. In this work, we do not design a specific technique to distinguish hard samples but the proposed method can handle hard samples well. In the early learning phase, both mislabeled samples and hard samples have lower weights compared to the easy samples, which forces the model to learn only from easy samples due to gradient scaling (Theorem 4.1). After the model has learned from easy samples for a while, the model then gradually learns from hard samples rather than the mislabeled samples. This is because the hard samples are more connected to the easy samples than the mislabeled samples, as the former two share some common features. Therefore, the proposed method can continuously learn from clean samples (both easy and hard) and prevent the memorization of mislabeled samples. In Figure 2, we can observe less overlap between clean and mislabeled samples compared to using the loss distribution [1]. Studying hard samples is an important task in noisy label scenarios. Similar procedures have been proposed to distinguish the hard samples in [2,3], i.e., training the classifiers only on easy samples, then using the classifiers to further distinguish the hard samples from the mislabeled samples.
>
> **Q2. Some clean samples are given low weights, and some mislabeled samples are given high weights.**
> **A2.** First, the clean samples given low weights are actually hard (clean) samples. As we discussed in A1, a classifier trained only on easy samples is able to discriminate hard samples from mislabeled samples. Since CIFAR-10 with 40% noise contain more easy samples than 60% noise, the classifier trained on 40% noise has stronger capability to discriminate hard samples than the classifier trained on 60% noise. Therefore, in Figure 2, we can observe that in CIFAR-10 with 40% noise, less clean samples have lower weights compared to 60% noise. Second, for all noisy cases in Figure 2, there exist mislabeled samples given high weights. We investigate some samples and find that these mislabeled samples are hard samples with "proper" wrong labels. For example, given a hard sample $x_{\text{h}}$: a cat that looks like a dog due to blurry resolution (e.g. 32 $\times$ 32). When generating simulated label noise, it is likely to assign a dog label for $x_{\text{h}}$. Compared to other mislabeled samples, this kind of mislabeled sample can be learned easily by the classifier, resulting in a high weight.
>
> **Q3. Combining NAL with MixUp.**
> **A3.** MixUp can be regarded as a strong augmentation and regularization technique. Theoretically, combining NAL with MixUp does not guarantee performance gains. Empirically, our experiments show performance improvements when using MixUp, especially at a high noise level. Below are the results on CIFAR-10.
> | Methods | sym 40% | sym 80% |
> |:-:|:-:|:-:|
> | NAL (ours) | 93.49 ± 0.07 | 80.98 ± 0.27 |
> | NAL (ours) + Mixup | 93.85 ± 0.11 | 83.71 ± 0.52 |
>
> (Note that the reason we do not use MixUp is for fair comparison with other methods that only modify the loss.)
>
> **Q4. Comparison with FINE and PES.**
> **A4.** The results are collected by running their official code and the backbone is ResNet34 for consistency.
>
> | Datasets | Methods | sym 20% | sym 50% | sym 80% | asym 40% |
> |:-:|:-:|:-:|:-:|:-:|:-:|
> | CIFAR-10 | FINE | 91.0 ± 0.1 | 87.3 ± 0.2 | 69.4 ± 1.1 | 89.5 ± 0.1 |
> | CIFAR-10 | PES | 92.3 ± 0.3 | 86.5 ± 0.5 | 28.0 ± 2.7 | 89.9 ± 0.6 |
> | CIFAR-10 | NAL (ours) | 94.4 ± 0.0 | 91.9 ± 0.1 | 81.0 ± 0.3 | 92.1 ± 0.1 |
> | CIFAR-100 | FINE | 70.3 ± 0.2 | 64.2 ± 0.5 | 25.6 ± 1.2 | 61.7 ± 1.0 |
> | CIFAR-100 | PES | 68.9 ± 0.5 | 58.9 ± 2.7 | 15.4 ± 3.5 | 63.3 ± 1.2 |
> | CIFAR-100 | NAL (ours) | 77.8 ± 0.3 | 70.1 ± 0.3 | 36.8 ± 0.7 | 74.7 ± 0.1 |
>
> **Q5. Performance on instance-dependent noise.**
> **A5.** We evaluate the proposed method using PMD instance-dependent noise from PLC [4]. For consistency with the PLC, the backbone is PreActResNet34.
> | Datasets | Methods | Type I (35%) |  Type II (35%) | Type III (35%) |
> |:-:|:-:|:-:|:-:|:-:|
> | CIFAR-10 | PLC | 82.80 ± 0.27 | 81.54 ± 0.47 | 81.50 ± 0.50 |
> | CIFAR-10 | NAL (ours) | 88.81 ± 0.13 | 87.66 ± 0.23 | 88.57 ± 0.16|
> | CIFAR-100 | PLC | 60.01 ± 0.43 | 63.68 ± 0.29 |  63.68 ± 0.29 |
> | CIFAR-100 | NAL (ours) | 66.55 ± 0.16| 67.15 ± 0.11| 66.59 ± 0.40|
>
> [1] Arazo, Eric, et al. "Unsupervised label noise modeling and loss correction." ICML 2019.
> [2] Bai, Yingbin, et al. "Me-momentum: Extracting hard confident examples from noisily labeled data." ICCV 2021.
> [3] Cordeiro, Filipe R., et al. "PropMix: Hard Sample Filtering and Proportional MixUp for Learning with Noisy Labels." BMVC 2021.
> [4] Zhang, Yikai, et al. "Learning with feature-dependent label noise: A progressive approach." ICLR 2021.

---

### Official Review · Reviewer_umW7 · 2022-07-09

**Rating:** 4
**Confidence:** 4
**Soundness:** 2 fair
**Presentation:** 2 fair
**Contribution:** 2 fair

**Summary:**

This paper presents a method for training models with noisy labels. The proposed method learns a weight for each training sample based on its noise level (measured by the difference between model prediction and label), using an attention branch. Experimental results on benchmark datasets show that the proposed method was able to improve the model performance compared to several noisy training approaches.

**Questions:**

1. All the proposed techniques in this paper are built based upon the observation that noisy samples have higher loss compared to clean samples. How does the proposed method handle samples with different difficulty levels?
2. It is unclear if the noise attention loss is used for the model training alone or in conjunction with the cross entropy loss.
3. Math notation needs to be improved. For example, $j$ is never defined in Sec 3.3 and it is used represent different things in Lemma 4.1 and Theorem 4.1. What is the difference between $t$ and $\hat{t}$ in Algorithm 1?


**Ethics Review Area:**

["I don’t know"]

**Strengths And Weaknesses:**

__Strengths__
- The idea is simple and experimental results are promising.

__Weaknesses__
- The theoretical justification is weak. Lemma 4.1 is essentially saying noisy labels hurts the model that trained with cross entropy loss. This is a well-known theory. I don't see how make it as a Lemma helps the discussion in this paper.

---

> ### Author Response · Authors · 2022-08-02
> **Response to Reviewer umW7**
>
> Thank you for comments and questions. We address them below.
>
> **Q1.** **The purpose of Lemma 4.1.**
> **A1.** Lemma 4.1 is used to better explain the gradient scaling effect in Theorem 4.1.
>
> **Q2.** **How does the proposed method handle samples with different difficulty levels?**
> **A2.** We assume the training set consists of mislabeled samples, easy (clean) samples and hard (clean) samples. In the early learning phase, both mislabeled samples and hard samples have lower weights compared to the easy samples, which forces the model to learn only from easy samples due to gradient scaling (Theorem 4.1). After the model has learned from easy samples for a while, the model then learns from the hard samples rather the mislabeled samples. This is because the hard samples are more connected to the easy samples than the mislabeled samples, as the former two share some common features. Therefore, the proposed method can continuously learn from clean samples (both easy and hard) and prevent the memorization of mislabeled samples. In Figure 2, we can observe that our weight distribution has less overlap between clean and mislabeled samples compared to directly using the loss distribution [1].
>
> **Q3.** **Is the noise attention loss used for the model training alone or in conjunction with the cross entropy loss?**
> **A3.** We use the noise attention loss alone in model training, as shown in Algorithm 1.
>
> **Q4.** **About notations.**
> **A4.** 1) The $j$ in Sec 3.3 indicates the $j$-th epoch, which is from epoch 1 to epoch $m$. 2) The $j$ in Lemma 4.1 represents $j$-th entry of the prediction (i.e. $p_{j}$) or the ground truth distribution (i.e. $q_{j}$ ) or the logits (i.e., $z_{j}$). 3) $\hat{t}$ is a typo, we only have $t$ as defined in Sec 3.3. We will correct it and make the notation clearer in the new version.
>
> [1] Arazo, Eric, et al. "Unsupervised label noise modeling and loss correction." ICML 2019.

---

### Official Review · Reviewer_fZY5 · 2022-07-12

**Rating:** 6
**Confidence:** 5
**Soundness:** 3 good
**Presentation:** 3 good
**Contribution:** 3 good

**Summary:**

This paper suggests attention-based method to distinguish mislabeled data and decrease their impact during training. Specifically, the authors add an attention branch at the end of the feature-extraction layer (a.k.a. after the penultimate layer) that outputs a confidence score $\tau$. Then, the attention layer's confidence score $\tau$ is incorporated into the loss function to softly divide samples into clean ones and mislabeled ones; the gradient of clean ones is trained as-is due to their early learning phenomenon, while that of mislabeled ones is not trained because $\tau$ is also trained to decrease the impact of mislabeled ones. The paper supports their idea both theoretically and empirically.

**Questions:**

The questions would be about how to deal with the above weakness points.

**Limitations:**

No critical limitation is found up to this point.

**Strengths And Weaknesses:**

## Strength

The strength of this paper can be elaborated as follows:
1. The writing is easy and clear to follow.
2. Incorporation of an attention layer and temporal ensembling-based target estimation is well-motivated and harmonious.
3. Theoretical justification and empirical support are sound.

## Weakness

1. A strong weakness is the use of two hyperparameters $\lambda$ and $\alpha$. Especially, $\lambda$ should be manually decided for each dataset, which is burdensome and hence becomes a strong hurdle for the suggested method.
2. The baselines in the experiments are inconsistent, possibly due to the copy of experimental results from original papers.
3. Why accuracy results on some of the two CIFAR datasets, Clothing 1M, and Webvision do not have variance results? They should be consistently reported.

---

> ### Author Response · Authors · 2022-08-01
> **Response to Reviewer fZY5**
>
> Thank you for a thoughtful review and questions. Please see below for answers to your questions.
>
> **Q1.** **Selection of hyperparameters.**
> **A1.** First, the sensitivity to $\alpha$ is quite mild as shown in Figure 4(f). We use a fixed $\alpha=0.9$ for all datasets. As for $\lambda$, its optimal value does depend on the complexity of the dataset. In Figure 4(e), the accuracy results using very small $\lambda$s (i.e. 0, 0.01, and 0.05) are only to demonstrate the effect of boost term $\mathcal{L}_{\text{b}}$. In our experiments, we perform a grid search of $\lambda$ from [0.1, 0.5, 5, 10, 50] using a noisy validation set sampled from the noisy training set (Note that the reason why a noisy validation set works has been empirically explored in [1] and theoretically proved in [2]). Therefore, we can still easily find the optimal $\lambda$ for different datasets, since it is the only hyperparameter that needs to be tuned.
>
> **Q2.** **Inconsistent baselines for different noises**.
> **A2.** The datasets and noise assumptions used for evaluation vary in papers. The approaches (e.g. NLNL [3] and DAC [4]) that simply modify the training loss are usually evaluated on simulated label noise, while only recently proposed methods (e.g. ELR [1], FINE [5], and Nested [6]) have been further evaluated on one or two real-world noisy datasets. In contrast, our experiments broadly cover three real-world noisy datasets to demonstrate that the proposed method can provide substantial improvements in different label noises. Therefore, you may observe inconsistencies in the baselines of different datasets in our paper, which also commonly happens in existing works [1,5,6].
>
> **Q3.** **Some accuracy results have variance while some don't**.
> **A3.** Generally, whether the result for a certain dataset have variance depends on two conditions: i) The noisy labels are fixed; ii) A validation set is provided. Only if both conditions are met, the result can have no variance. According to the dataset information, we have the following table:
> | Datasets/Conditions | Fixed noisy labels? | Validation set is provided? | Results have variance? |
> |:--:|:-:|:-:|:-:|
> | CIFAR with simulated noise | No | No | Yes |
> | Animal-10N | Yes | No | Yes |
> | Clothing1M | Yes | Yes | No |
> | Webvision  | Yes | Yes | No |
>
> In our experiments, randomly splitting a noisy validation set or randomly generating label noise requires multiple runs to obtain the average results and its standard deviation. Therefore, the results on CIFAR and Animal-10N have variance, while the results on Clothing1M and Webvision usually don't have variance, which is consistent with most existing works [1,5,6]. Note that some of the two CIFAR results are taken from their original papers, including Joint Opt, NLNL, DAC and SAT [7]. Because the results in their original papers do not have variance, their results in Table 1 also have no variance. The reason we take the results directly from the original paper is that either the paper does not provide the code, or the provided code could not achieve the results reported in the original paper. For example, we run the official code of SAT, the results on CIFAR-10 are below:
>
> | Source | sym 20% | sym 40% | sym 60% | sym 80% |
> |:-:|:-:|:-:|:-:|:-:|
> | From original paper | 94.14 | 92.64 |  89.23 | 78.58 |
> | Our re-run results | 93.92 ± 0.07 | 92.56 ± 0.23 | 89.14 ± 0.25 | 74.84 ± 1.92 |
>
> It can be observed that on CIFAR-10 with sym 80% noise, our re-run results are much lower than the reported results. Therefore, we chose to take the results directly from the original papers.
>
> [1] Liu, Sheng, et al. "Early-learning regularization prevents memorization of noisy labels." NeurIPS 2020.
> [2] Chen, Pengfei, et al. "Robustness of accuracy metric and its inspirations in learning with noisy labels." AAAI 2021.
> [3] Kim, Youngdong, et al. "Nlnl: Negative learning for noisy labels." ICCV 2019.
> [4] Thulasidasan, Sunil, et al. "Combating Label Noise in Deep Learning using Abstention." ICML 2019.
> [5] Kim, Taehyeon, et al. "Fine samples for learning with noisy labels." NeurIPS 2021.
> [6] Chen, Yingyi, et al. "Boosting co-teaching with compression regularization for label noise." CVPR 2021.
> [7] Huang, Lang, et al. "Self-adaptive training: beyond empirical risk minimization." NeurIPS 2020.

---

> > ### Comment · Reviewer_fZY5 · 2022-08-09
> > **Response to the authors**
> >
> > I appreciate the authors' response to my questions which would help readers' understanding of the paper. Having read all the responses to my and other reviewers' review, there is no further question I would like to ask except for an additional experiment result on hard samples (say, for clean samples that contain high loss in the early learning phase, how does NAL work during training?)
> >
> > In this regard, I would like to keep my review score up to now.

---

> > > ### Author Response · Authors · 2022-08-09
> > > **Thank you for the response**
> > >
> > > Thank you for the response.  We have updated the Appendix for more experiments on hard samples. In Appendix F, we provide the movement of $\tau$ distribution in the early learning stage to empirically explain how hard samples can be learned by using the proposed method.

---

### Author Response · Authors · 2022-08-09
**Clarification of Our Contribution and Response**

Thanks for the constructive comments from the reviewers!

Our major contribution is to use an extra branch and propose a specific loss function to learn weights for training samples. The designed loss scales the gradients of the samples according to the learned weights, thus preventing the memorization of noisy labels in training. However, the significance of this contribution, as recognized by Reviewer fZY5 and BcSR, may be a little bit underrated by Reviewer umW7.

The reviewers' questions can be summarized as follows:
1. Selection of hyperparameters.
2. Evaluation details.
3. Discussion on hard samples.
4. Comparison with other related methods and instance-dependent noise.

We have answered the above points in our response. Please take a look at them.

Regards,

---

### Meta-Review · Area_Chair_49HN · 2022-08-31

**Recommendation:** Accept
**Confidence:** Less certain

**Metareview:**

The work proposes a simple method for training an 'attention layer' that can give weights for different input samples. These weights are learned during the training process. This method appears to the theoretically justified, the method relatively simple and the empirical results seem reasonable. One concern that I share with the reviewers is about the hard (but correctly labeled) samples. The authors provide some explanation and results in the appendix, which mostly allay the concerns. I have to agree that introducing a somewhat sensitive parameter $\lambda$ is not ideal, but overall the empirical and theoretical justifications tip the balance towards acceptance in this case.

**Award:**

No

---

### Decision · Program_Chairs · 2022-09-14

Accept